# Effects of Resilience Training on Resilient Functioning in Chronic Stress Situations among Cadets of the Swiss Armed Forces

**DOI:** 10.3390/healthcare11091329

**Published:** 2023-05-05

**Authors:** Madlaina Niederhauser, Regula Zueger, Hubert Annen, Nejla Gültekin, Zeno Stanga, Serge Brand, Dena Sadeghi-Bahmani

**Affiliations:** 1Military Academy, Swiss Federal Institute of Technology ETH Zurich, 8903 Birmensdorf, Switzerland; madlaina.niederhauser@gmail.com (M.N.); regula.zueger@milak.ethz.ch (R.Z.); 2Centre of Competence for Military and Disaster Medicine, Swiss Armed Forces, 3008 Bern, Switzerland; nejla.gueltekin@vtg.admin.ch (N.G.); zeno-giovanni.stanga@vtg.admin.ch (Z.S.); 3Center for Disaster Psychiatry and Disaster Psychology, Psychiatric Clinics of the University of Basel, 4002 Basel, Switzerland; serge.brand@upk.ch; 4Center for Affective, Stress and Sleep Disorders (ZASS), Psychiatric University Hospital Basel, 4002 Basel, Switzerland; 5Division of Sport Science and Psychosocial Health, Department of Sport, Exercise and Health, University of Basel, 4052 Basel, Switzerland; 6Sleep Disorders Research Center, Kermanshah University of Medical Sciences, Kermanshah 67146, Iran; bahmanid@stanford.edu; 7Substance Abuse Prevention Research Center, Health Institute, Kermanshah University of Medical Sciences, Kermanshah 67146, Iran; 8School of Medicine, Tehran University of Medical Sciences, Tehran 25529, Iran; 9Center for Disaster Psychiatry and Disaster Psychology, Psychiatric University Hospital Basel, 4002 Basel, Switzerland; 10Department of Psychology, Stanford University, Stanford, CA 94305, USA

**Keywords:** resilience training, resilient functioning, resilience score, chronic stress

## Abstract

Research on resilient functioning has gained increasing interest, and some recent studies interpreted resilience in the sense of resilient functioning to stress. In the present study, we investigated the associations between resilient functioning and coping strategies, stress reactivity, self-efficacy, and well-being, and we examined whether resilient functioning could be improved through a training intervention. The participants were 110 male cadets from two infantry officers’ schools of the Swiss Armed Forces. The schools were divided into an intervention and control group. The participants in the intervention group took part in the resilience training intervention, whereas the participants in the control group performed military training as usual. Data were assessed before and after the intervention period. Results showed that resilient functioning was positively associated with task-oriented coping and well-being and negatively associated with emotion-oriented coping and stress reactivity. Furthermore, resilient functioning significantly improved in the intervention group from pre- to post-intervention. The results suggested that specific interventions have the power to increase resilient functioning.

## 1. Introduction

Resilience is defined as the ability of a person to retain their mental health despite psychological adversity [1,2]. Resilience results from various individual protective cognitive, affective, behavioral, and social factors [3], and such protective factors influence how a person reacts to adversity [4]. For instance, important resilience factors include higher scores for self-regulation [5], self-efficacy [6,7], optimism, favorable coping strategies [8,9,10], better stress reactivity [11], more positive affect, higher behavioral control [12], and better cognitive emotion regulation strategies [13].

In the past, resilience trainings were designed to increase the resilience of different participant samples, such as healthy adults, military personnel, and employees from private companies (for an overview, see [8,10]). Results showed that such specific resilience trainings could enhance problem-solving and coping strategies [14,15], decrease ineffective coping strategies [16], improve positive emotions, mindfulness, and acceptance [17], and create a more positive state of mind [18]. Further, a typical way to verify the positive effects of resilience training is to examine changes in maladaptive behavioral outcomes, including depressive symptoms [19,20], anxiety [21,22], anger [23], distress, poor general health [24], social strain [25], fatigue, and sleep difficulty [23,26].

These results mentioned previously suggested that resilience training interventions were successful and participants became more resilient. However, this interpretation might bear two limitations.

First, evidence regarding an increase in one specific resilience factor does not causally imply that the participants have become more resilient in both the general short-term and the longer-term; more specifically, only the respective resilience factors have been improved. Whether these factors lead to increased resilience in adversities over time could not be tested. Further, recording mental health outcomes without including the actual adversity of a person only provides information about the mental health of this person and does not provide information on changes in their resilience [27,28].

Second, for an accurate determination of resilience, an individual has to be exposed to a challenging context because a person can only prove individual resilience in cases of necessity and exposure to stress [8,10]. In this view, little research has investigated the impact of resilience training interventions in acute [29,30,31] or chronic stress situations [20,22]. In addition, and based on Lazarus and Folkman’s [32] seminal work of the cognitive appraisal of stress, the way individuals perceive stress and stressors is highly individualized [27,32], i.e., an identical situation interpreted as threatening to one person might be perceived as trivial to another. This observation holds true even in a multi-stressor environment like military training in which military personnel have an increased risk for adversity [33,34,35] and objective stress [20,36,37,38]. Such stress results from a highly structured environment with lack of personal autonomy [15,37,38] combined with stressors like sleep deprivation, environmental challenges, and psychological strain [37,39]. While such stressors are daily, their perception and cognitive–emotional elaboration underlies individual standards. As such, it appears plausible that individually perceived and appraised stress levels should be considered conceptually when comparing resilience training outcomes within a given sample. Given this, we claimed that assessing and measuring resilience is a methodological challenge.

While self-rating questionnaires are the gold standard to assess resilience more directly, it appears that such questionnaires prevalently capture resilience as a trait (e.g., Resilience Scale [RS-11]; [40]), as an assembly collection of resources and protective factors (e.g., Connor–Davidson Resilience Scale [CD-RISC]; [41]), or as an individual skill to recover from stress despite adverse circumstances (Brief Resilience Scale [BRS]; [42]). Previous studies using these resilience measurements showed that resilience could be increased through resilience interventions [43,44,45]. However, a key characteristic of the three questionnaires is that they refer to hypothetical situations (e.g., item one of RS-11: “When I have plans, I follow them”) or to universal statements (e.g., item four of BRS: “It is hard for me to snap back when something bad happens”) while apparently not considering the participant’s individual and specific actual stress situation. However, as mentioned above, to determine resilience, it is necessary for the individual to be exposed to stress in order to prove an individual’s resilience accurately [8,10]. Given this, the bottom line should be to test the effectiveness of a resilience intervention so participants actually become more resilient during individual stressful times and can function in a resilient fashion in a specific situation and given context.

With this in mind, the methodological challenge is to figure out the possibility of both quantifying resilience more directly—not only with relying on indirect indicators of resilience factors or mental health—and to include individual stress. To this end, a validated approach is to calculate the so-called Stress Resilience or Resilient Functioning Score (RFS; [46,47]), which quantifies resilience while considering a participant’s individual and specific stress situation. This statistical method for the operationalization of resilience in stress-exposed individuals uses a regressive correlation of stress and mental health variables in a specific sample as a benchmark for the average response to the stressor. An individual’s deviation from this regressive correlation is then interpreted as follows: an individual’s deviation below the averaged regressive correlation indicates that this individual is more resilient, relative to a specific sample, while an individual’s deviation above the averaged regressive correlation indicates that this individual is less resilient, relative to a specific sample. For a better illustration and understanding of this score, Figure 1 shows an example of a positive relationship between stress and symptoms of depression, as frequently documented in the literature [6,48]. A closer look at this example shows that there are individuals who develop few symptoms of depression even under high chronic stress (green dots in Figure 1). These individuals are called resilient and function resilient in chronic stress situations because they stay healthy [1,2] and remain functionally stable [49,50] despite exposure to chronic stress. Other individuals, however, develop more depressive symptoms under chronic stressful circumstances (red dots in Figure 1). These individuals have low resilient functioning in a specific stress situation. Resilience, or resilient functioning, is thus operationalized as the ratio of individual’s health variables and experienced stress related to the same and specific sample. Therefore, the Resilient Functioning Score enables the measurement of resilient functioning at a given point as an index of resilience [7,46]. While a Resilient Functioning Score (RFS) is not generalizable and comparable with resilient scores derived from other samples, the RFS serves as a benchmark within the specific sample and context. Thus, resilience can be objectivized through the Resilient Functioning Score (RFS), which reflects the actual resilient functioning [7]. As a further result, compared to questionnaire-based resilient scores, the RFS assesses resilience more individually, adequately, directly, and in relation to a specific sample and context. In sum, the idea of this approach is to measure resilience as a higher or lower level of resilient functioning in stressful situations in the sense of successful adjustment to stress [51,52].

Unlike conventional self-rating questionnaires that assess resilience traits, the advantages of measuring resilient functioning with the RFS have led to the following observations: studies with adolescents showed that friendship was a positive predictor for resilient functioning [46]; further, friendship quality and resilient functioning after childhood and adolescence adversity were interrelated [53]. A study focusing on resilient functioning trajectories in university students during the first COVID-19 pandemic lockdown in Germany reported that resilient functioning trajectories remained fairly stable over time and that more self-care was associated with a higher resilient functioning trajectory [54]. Another study used resilient functioning scores as a base for testing differences in self-efficacy with network analyses. High resilient functioning adults had a higher and stronger connectivity in self-efficacy than low resilient functioning adults [7]. Further, the RFS was also used in theoretical conceptualizations to explain resilient functioning after childhood maltreatment [55] or to establish a framework for neurobiological studies on resilience [56]. These studies convincingly reported that the RFS is appropriate for measuring resilient functioning.

Given this background, it is important to study resilient functioning in chronic stress situations and to investigate the possibility of increasing resilience through a specific resilience training intervention. In the same vein, it appears that research on the associations between the RFS and well-known resilience factors, such as coping strategies [8], self-efficacy [7], and stress reactivity [11], are missing so far. Accordingly, the objectives of the present study were to explore:Whether RFS was related to well-established resilience factors, such as coping, self-efficacy, and stress reactivity;Whether RFS was related to well-being in chronic stress situations; andWhether resilient functioning could be improved through participation in resilience training.

We formulated three hypotheses. First, following previous studies investigating resilience with questionnaires, we hypothesized that RFS scores will be correlated with higher scores regarding favorable coping strategies [8] and higher self-efficacy [7] and lower stress reactivity [11]. Second, based on previous studies on resilience and health outcomes, we hypothesized that the RFS will be correlated with higher well-being [57,58]. Third, based on the fact that resilience trainings were efficient for improving resilience factors and mental health outcomes [8,10], we hypothesized that resilient functioning measured with the RFS can be improved through a resilience training.

## 2. Methods

### 2.1. Participants and Study Design

This intervention study was conducted as part of a research project on aspiring military officers of the Swiss Armed Forces. In Switzerland, military service for men is compulsory. After basic training, the most skilled recruits are voluntarily recruited for continuation as cadre. After training in the non-commissioned officers’ school, fifteen weeks of officers’ school (OS) starts, which is considered very physically and mentally stressful and demanding. The cadets are pushed to their limits every day during the extraordinarily intensive training.

The study participants were 161 cadets from two Swiss Armed Forces infantry officer training schools in 2016 and 2017. The cadets were briefed before the study, and they voluntarily signed a written consent form. Participation or non-participation in the present study had no impact on a cadet’s current and future military career. Each officers’ school was assigned either as an intervention group (IG) or as a control group (CG) to avoid cross-contamination of training content [59]. The participants in the control group performed the OS as usual without any intervention (*n_CG_* = 88); in contrast, the intervention group participated in the resilience training intervention (see description below) during the OS (*n_IG_* = 73).

The study was carried out according to the current and seventh revision of the Ethical Principles of the Declaration of Helsinki for experiments involving human subjects [60] and reviewed by the Ethics Committee Zurich (Zürich, Switzerland; “Kantonale Ethikkommission KEK”; Req-2016-00465).

### 2.2. Resilience Training Intervention

The resilience training consisted of four sessions and was implemented weekly between the fourth and seventh week of the OS, and each session lasted 90 min. Between the sessions, the participants practiced the acquired skills in military everyday life and completed specific homework. Experienced instructors in teaching resilience performed each single session of the training focusing on different thematic priorities for enhancing participants’ resilience and resilient functioning. Each session included short theories instructed by the project team and then individual practical exercises in moderated groups. Moderators supported participants in their improvements and self-reflections and moderated the discussions to further improve participants’ learning and skill acquisition. The contents of the sessions were based on cognitive behavioral theories and on positive psychology interventions and were as follows:Week One: Participants learned to consider their individual emotions, thoughts, and behaviors, reflect on their contributions to interpreting stress [61], detect individual thinking traps [9], and improve their optimism [61].Week Two: Participants learned to identify their values and core beliefs [9].Week Three: Participants learned to identify their individual coping strategies [32,59], modify them in cases of dysfunctionality, and optimize their performance [62,63,64,65].Week Four: Participants learned to distinguish different communication styles, detect favorable ones, and detect their individual character strengths [66,67,68]. As a final exercise, a “bombardment of strengths” was made within the small groups to enhance positive emotions [69].

### 2.3. Measurement Scales

Sociodemographic and pre-measurement variables were collected after signing consent forms (in week 1) and before the resilience training intervention. The post-measurements were assessed after the end of the intervention (in week 7).

#### 2.3.1. Chronic Stress

We assessed chronic stress with the perceived stress questionnaire (PSQ; [70]). The PSQ assesses subjective experiences of stressful situations in the previous four weeks (e.g., “I have too many things to do”). Each item was scored on a four-point Likert scale ranging from rarely (=1) to usually (=4). Higher sum scores reflect higher subjectively perceived chronic stress. The current sample had a satisfactory consistency (α = 0.84).

#### 2.3.2. Vital Exhaustion

The short version of the Maastricht VE Questionnaire (MQ; [71]) was used to measure vital exhaustion. Each item (e.g., “Do you often feel tired?”) was rated on a scale ranging from no (=0) to uncertain (=1) to yes (=2). Higher sum scores reflect a higher vital exhaustion. The internal consistency was acceptable (Cronbach’s α = 0.77).

#### 2.3.3. Symptoms of Depression

The participants reported their symptoms of depression with the general depression scale (German version, ADS; [72]). The ADS consists of twenty items (e.g., “During the last week, I was depressed/dejected”) reflecting different symptoms of depression. Answers are given on a four-point Likert scale ranging from “rarely or not at all” (=0) to “usually, all the time” (=3). Higher sum scores reflect higher symptoms of depression. The internal consistency of the ADS was acceptable (Cronbach’s α = 0.71).

#### 2.3.4. Irritation

Irritation (I) was measured with the Irritation Scale [73]. Eight items (e.g., “I get irritated easily, although I don’t want this to happen”) assess emotional and cognitive irritation on a seven-point Likert scale ranging from 1 (=not at all) to 7 (=almost completely true). Higher sum scores reflect a higher irritation. The internal consistency of the irritation scale was good (Cronbach’s α = 0.82).

#### 2.3.5. Coping Strategies

Coping strategies were assessed with the German version of the Coping Inventory for Stressful Situations (CISS; [74]). Twenty-four items collected typical ways of dealing with stress, and answers are given on five-point scales ranging from 1 (=very untypical) to 5 (=very typical). Three subscales were calculated: (1) task-oriented coping, which reflects strategies to resolve the problem and cause of stress (e.g., “I work out a solution plan and execute it”), (2) emotion-oriented coping, which reflects negative emotional strategies reacting to stress (e.g., “I blame myself for getting into this situation”), and (3) avoidance-oriented coping, which includes avoidance tendencies in stress (e.g., “I go shopping”). Higher sum scores reflect higher coping tendencies. The internal consistency for the current sample was between acceptable and good for the subscales (between Cronbach’s α = 0.77 and α = 0.82).

#### 2.3.6. Self-Efficacy

Self-efficacy was assessed with the Questionnaire on Competence and Control Beliefs (FKK; [75]). Sixteen items (e.g., “I can determine a lot of what happens in my life myself”) assess internal and external control beliefs on six-point Likert scales ranging from 1 (=very false) to 6 (=very true). Higher sum scores reflect higher self-efficacy. The internal consistency of the current sample was acceptable (Cronbach’s α = 0.75).

#### 2.3.7. Stress Reactivity

Stress reactivity was measured with the Stress Reactivity Scale (SRS; [76]) by summing up all the items. Each of them consists of two parts. The start of a sentence describes a stressful situation (e.g., “When I have conflicts with others that may not be immediately resolved…”), and the second part includes three options with different stress responses (1 = “I generally shrug it off”, 2 = “It usually affects me a little”, and 3 = “It usually affects me a lot”). Higher sum scores reflect higher stress reactivity. The internal consistency of the current sample was good (Cronbach’s α = 0.79).

#### 2.3.8. Psychological Well-Being

We assessed psychological well-being with the World Health Organization Well-Being Index (WHO-5; [77]). Five items (e.g., “I have felt cheerful and in good spirits”) with a scale from 0 (=at no time) to 5 (=all the time) measure mental well-being over the last two weeks. Higher scores reflect higher well-being. The current sample had a good consistency (Cronbach’s α = 0.83).

### 2.4. Analytical Plan

In a first step, resilient functioning scores were calculated as indicators of resilience with established procedures described in previous publications [7,46,47]. To this end, we first performed a principal component analysis (PCA) to obtain single indicators for stress and mental health. The stress indicator was thereby represented with the standardized sum scores of the five subscales of the perceived stress questionnaire (PSQ). The mental health indicator was calculated with inverted sum scores of the symptoms of depression (ADS), vital exhaustion (MQ), and irritation (I). Next, partial least squares regressions (PLSR; [78]) were calculated with the indicators of stress and mental health to extract regression residuals. This extracted residuals were interpreted as resilient functioning scores [7,47]. Positive values reflect a high resilient functioning at a certain time point, whereas negative values reflect low resilient functioning at a certain time point. All further statistical evaluations listed below regarding resilient functioning are based on these RFSs produced by the PCA.

Second, with Pearson’s correlations, we identified the associations between resilient functioning and coping strategies, self-efficacy, stress reactivity, and well-being.

Third, an ANOVA for repeated measures was used to detect differences in pre- and post-test measurement between the two groups. The integrated factors were Time (pre- and post-test), Group (intervention vs. control group), and the Time × Group interaction. A Levene’s test was used to test the homogeneity of error variances, and a Box test was used to assess the homogeneity of covariance.

Fourth, we calculated differences in pre- and post-test results between the two groups with single *t*-tests.

Fifth, individual changes over time in resilient functioning were calculated with the reliable change index (RCI; [79]. With the RCI, participants who had significant individual changes from the pre- to post-test could be identified. Significant differences between the IG and CG in RCI were calculated with Chi-square tests.

Partial eta-squared [*η_p_*^2^]) were used to report effect sizes for the F-statistics and interpreted according to Cohen (1988), ranging from trivial to large effects [80,81].

SPSS^®^ 28.0 (IBM Corporation, Armonk, NY, USA) for Windows was used to perform all analysis, and the level of significance was set at *p* < 0.05 for all tests.

## 3. Results

### 3.1. Characteristics of Participants

A total of 161 cadets started the OS in the study period. Female participants were excluded from the present study analysis because there were only four female participants in the control group. Fifteen participants quit the officers’ school because of medical or military reasons. Furthermore, some participants were excluded because of unreliable answers (*n* = 7; measured with the ADS lie subscale) and missing data (*n* = 25). Slightly higher scores in chronic stress were found in participants who did not complete the OS (*M* = 2.06, *SD* = 0.41; *t*(16.58) = −2.23, *p* = 0.04, *d* = 0.67) than those who completed the OS (*M* = 1.81, *SD* = 0.33). These results are not surprising because the OS is a highly stressful time; accordingly, it can be assumed that less resilient cadets were not able to persevere. We continued the statistical analysis because there were no significant differences at pre-test in vital exhaustion, symptoms of depression, or irritation, nor in the number of dropouts between the intervention group (IG) and the control group (CG).

The remaining 110 participants had a mean age of 21 years (M = 20.94, SD = 1.64). There were no significant differences in age or education between the intervention and the control groups. Table 1 reports the participants’ age and highest educational level separately for the intervention and control conditions.

### 3.2. Correlations of Resilient Functioning with Coping, Self-Efficacy, Stress Reactivity, and Well-Being

Table 2 (pre-test) and Table 3 (post-test) report the descriptive statistical indices and the Pearson’s correlation coefficients between resilient functioning scores and scores for coping, self-efficacy, stress reactivity, and well-being. At pre-test, higher scores in resilient functioning were associated with lower symptoms in emotion-oriented coping and stress reactivity and with higher scores in well-being (always significant *p*-values). Further, descriptively, higher scores in resilient functioning were associated with higher scores in task-oriented coping and self-efficacy. At post-test, higher resilient functioning scores were significantly associated with lower scores in emotion-oriented coping and stress reactivity and with higher scores in task-oriented coping. Further, descriptively, higher scores in resilient functioning were associated with higher scores in well-being.

### 3.3. Improvement of Resilient Functioning from Pre- to Post-Test between and within the Intervention and Control Conditions

To test whether resilience training could improve resilient functioning, a repeated measure design with three factors was performed: Time (pre- and post-test), Group (IG vs. CG), and the Time × Group interaction. The results revealed significantly different progress (*F*(1, 108) = 6.114, *p* = 0.015, partial η^2^ = 0.054, small effect size). Results are shown in Figure 2. The repeated measure design for each group revealed a slight increase in resilient functioning in the intervention group (*F*(1, 50) = 3.432, *p* = 0.070, partial η^2^ = 0.064) but not in the control group. There were no significant differences between groups in the pre- or post-tests.

### 3.4. Reliable Change Index of Resilient Functioning

The number of individuals who demonstrated significant changes in resilient functioning was compared between the two groups (see Table 4). We found that more participants in the intervention group showed a significant increase in resilient functioning (*X*^2^(2, *N* = 110) = 6.414, *p* = 0.022) compared to participants in the control group. Furthermore, we found no significant differences between decreases in resilient functioning.

## 4. Discussion

The aims of the present study were to investigate the resilient functioning score (RFS) as a measuring instrument for resilience in chronic stress situations and to evaluate the influence of resilience training on this resilient functioning score. Furthermore, the relationships between resilient functioning and coping strategies, stress reactivity, self-efficacy, and well-being were calculated. The key findings were that higher resilient functioning was associated with lower emotion- and higher task-oriented coping, lower stress reactivity, and higher scores for actual well-being. Moreover, resilience training increased resilient functioning. The present data add to the current literature in that resilient functioning—understood as behaving in a resilient fashion in chronic stress situations—can be improved through a specific and standardized resilient training intervention.

Three hypotheses were formulated, and each hypothesis is considered now in turn.

With the first hypothesis, we predicted that higher resilient functioning scores were correlated with resilience-related factors, such as higher task-related and lower emotion-related coping strategies, higher self-efficacy, and lower stress reactivity, and the data did generally confirm these predictions. These results were in line with the current literature, which found associations between higher traits of resilience with more favorable coping strategies [8] and stress reactivity [11].

However, and surprisingly, resilient functioning scores were statistically unrelated to scores for self-efficacy. Though speculative, we offer two explanations for this pattern of results. First, it is conceivable that the structure of the military context and military hierarchies offer little individual opportunity for feeling self-efficacious. Self-efficacy is defined as “people’s beliefs about their capabilities to produce designated levels of performance that exercise influence over events that affect their lives” [82]; therefore, it relates to the experience of control over a situation [83]. Thus, autonomy is important for self-efficacy, and playing a subordinate role in a situation or domain could have negative consequences for self-efficacy [84]. A recently published study, which found a positive correlation between the hierarchy level of leaders and self-reported self-efficacy, supports this assumption [85]. Second, the sample consisted of highly selected cadets of officers’ school. In particular, young, confident, and mentally stable adults pursue this military career. The high means and lower standard deviations (*M* = 72.60, *SD* = 6.76) of the present sample compared to an adult normative sample (*M* = 64.2, *SD* = 10.25; [75]) support this assumption. Such low standard deviations and thus rather small variances make it difficult to find statistically significant results, and this could explain the marginally non-significant results. As such, further research is needed to understand the relationship between resilient function and self-efficacy.

The second hypothesis predicted that higher resilient functioning was related to higher well-being in chronic stress situations, and data did confirm this assumption. This pattern of results was in line with previous studies, which showed higher well-being scores among individuals self-reporting higher scores for resilience [57,58] compared to individuals self-reporting lower scores for resilience. However, while previous studies [57,58] investigated associations of well-being with trait resilience, the novelty of the current study is that for the first time, the association of resilient functioning, understood as staying mentally healthy in chronic stress situations, with higher well-being was observed.

The third hypothesis predicted that resilient functioning could be improved through a resilience training. Results showed that the resilience scores in the two groups differed from the pre-test to the post-test. While the intervention group did descriptively increase in resilient functioning, the control group did not. The current results highlight the effect of this resilience training intervention [61] and point out that next to outcome variables, resilient functioning in chronic stress situations can be improved through specific and standardized interventions. To date, this is the first study showing an increase in resilient functioning through a training intervention. As such, we claim that resilience training interventions have the potential to help participants maintain mental health despite high chronic stress.

The effect sizes of the observed results were small. This is in line with previous studies, which found mixed results for the effectiveness of resilience training in military contexts, often with weak or no effects [8,18,86]. Such small effects could be biased by other factors, including training settings and delivery formats [8] or individual factors like motivation [61].

The present study measured resilient functioning in chronic stressful situations. In contrast to previous studies, resilience was not measured with questionnaires or indirectly with outcome variables; rather, it was measured as resilient functioning in relation to individual stress and mental health variables. Therefore, the novelty of the study is that it showed that resilient functioning, understood as behaving in a resilient fashion in chronic stress situations, was associated with higher task-oriented coping and well-being and with lower emotion-oriented coping and stress reactivity. Furthermore, data suggested that resilient functioning can be strengthened by a resilience training intervention.

Regarding missing data and the attrition rate of 10% [87,88], we noticed that such rates were consistent with attrition rates observed in earlier resilience training studies with stress-exposed samples, such as military cadre [88] and personnel of blue light organizations [87]. Further, the scores at the pre-test between the remainders and dropouts demanded particular attention as higher scores in chronic stress were found at the pre-test for those cadets who dropped out. It can be assumed that participants already scoring high for chronic stress at the beginning of military training had less resources to meet the high requirements during the strenuous physical and mental training of an officer school (OS); as a result, resigning from the OS appeared to be a plausible solution.

The present findings are of practical importance and add a further piece to the existing resilience literature. The results showed that resilient functioning can be improved with training interventions. Compared to similar studies in the field of resilience training, the advantage of the present study was that we fully relied on individual resilient functioning, including individual chronic stress and mental health problems. More specifically, participants in the intervention group acted more resilient than participants in the control group because the former reported lower scores for mental health symptoms in individually perceived chronic stress situations. Thus, the present study was the first to demonstrate resilient functioning as a direct measure of resilience in chronic stress situations following a training intervention. As such, the present study’s results contributed to improving the quality of resilience research and to the operationalization of resilience in stressor-exposed individuals.

There were limitations in the present study. First, the participants were healthy and highly selected young men attending a demanding military course which makes generalization of the pattern of results difficult. Second, the randomization between groups was made by the military organizations which could have led to a bias [89]. Third, all results based on self-reporting scales which could bear the risk of answering with less care [90] or with social desirability [91]. Fourth, further relevant and well-known resilience factors such as optimism or self-regulation were not investigated in the study [5].

Future research should examine additional resilience factors and their association with resilient functioning. It is also of interest to examine whether resilient functioning is related to military qualifications or performance. Further, follow-ups three to six months after the intervention would allow researchers to understand whether long-term impacts are produced by such a brief and tight resilience intervention and to what extent.

## 5. Conclusions

This study investigated the use of a Resilient Functioning Score, based on the “residual” method, as a new direct measurement of resilience. Resilient functioning is understood as behaving in a resilient fashion in chronic stress situations. Data suggested that resilience training intervention could improve resilient functioning in chronic stress situations and that resilient functioning was associated with higher task-oriented coping and lower stress reactivity and emotion-oriented coping.

## Figures and Tables

**Figure 1 healthcare-11-01329-f001:**
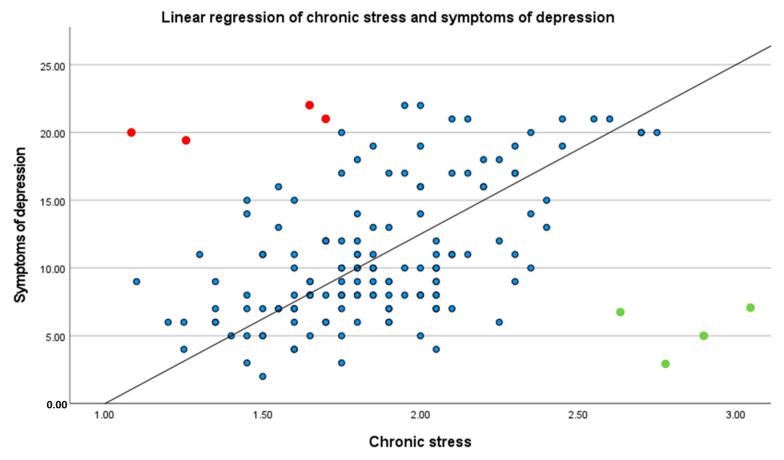
Example of a linear regression between chronic stress and symptoms of depression. Blue points illustrate the cloud of points of linear regression, some of these points were colored for a better understanding of the score: Green dots represent high resilient functioning individuals, and red dots represent low resilient functioning individuals.

**Figure 2 healthcare-11-01329-f002:**
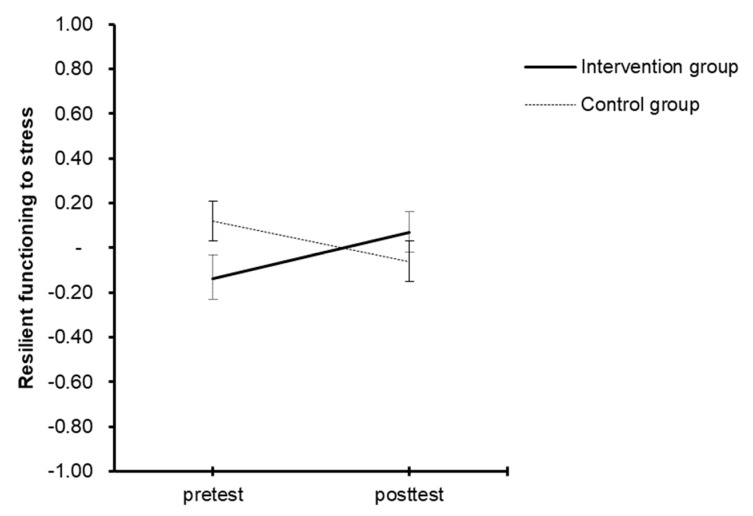
Mean scores in the pre- and post-tests for resilient functioning differentiated for the intervention and control groups.

**Table 1 healthcare-11-01329-t001:** Sample characteristics.

	Groups		Statistics
	Intervention	Control	
	*M* (*SD*)	*M* (*SD*)	
N	51	59	
Age (in years)	20.84 (1.42)	21.02 (1.81)	*t*(107.09) = −0.57, *p* = 0.580; *d* = 0.11)
Education level	*n* (%)	*n* (%)	*χ*^2^ (N = 110; *df* = 1) = 0.762, *p* = 0.383
Upper secondary school	98.0%	94.9%	
Tertiary level	2.0%	5.1%	

**Table 2 healthcare-11-01329-t002:** Correlations of resilient functioning with coping, self-efficacy, stress reactivity, and well-being at pre-test.

Variable	1	2	3	4	5	6	7	M (*SD*)
1. Resilient functioning								0 (0.72)
2. Task-oriented coping	0.17							3.93 (0.51)
3. Emotion-oriented coping	−0.38 ***	−0.46 ***						2.32 (0.61)
4. Avoidance-oriented coping	0.08	−0.08	0.08					3.10 (0.75)
5. Self-efficacy	0.18	0.53 ***	−0.60 ***	−0.01				72.60 (6.76)
6. Stress reactivity	−0.30 **	−0.43 ***	0.65 ***	−0.06	−0.52 ***			45.63 (6.29)
7. Well-being	0.23 *	0.15	0.32 ***	−0.04	0.32 ***	0.29 **		17.00 (4.12)

Note: *N* = 110, * *p* < 0.05, ** *p* < 0.01, and *** *p* < 0.001.

**Table 3 healthcare-11-01329-t003:** Correlations of resilient functioning with coping, self-efficacy, stress reactivity, and well-being at post-test.

Variable	1	2	3	4	5	6	7	M (*SD*)
1. Resilient functioning								0 (0.68)
2. Task-oriented coping	0.19 *							3.92 (0.57)
3. Emotion-oriented coping	−0.20 *	−0.41 ***						2.28 (0.63)
4. Avoidance-oriented coping	0.02	−0.12	0.10					2.88 (0.75)
5. Self-efficacy	0.13	0.51 ***	−0.52 ***	−0.15				72.65 (7.61)
6. Stress reactivity	−0.22 *	−0.39 ***	0.61 ***	−0.06	−0.49 ***			44.61 (7.28)
7. Well-being	0.19	0.33	0.24 **	−0.08	0.36 ***	0.37 **		16.09 (4.06)

Note: *N* = 110, * *p* < 0.05, ** *p* < 0.01, and *** *p* < 0.001.

**Table 4 healthcare-11-01329-t004:** Changes in the reliable change index of resilient functioning.

		IG (*n* = 51)	CG (*n* = 59)	Fisher’s Exact Test, Cramer’s V
Resilient functioning	Increase	8%	0%	*p* = 0.043, V = 0.209 [S]
	Decrease	0%	3%	*p* = 0.498, V = 0.127 [S]

Note: IG = intervention group, CG = control group, and [S] = small effect size.

## Data Availability

Data belong to the Swiss Armed Forces. Data are available to experts in the field upon request with a detailed description of the reason for the request, the precisely formulated hypotheses, along with the full description, where and how data are securely stored without being shared to third parties.

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
