# Peer review of "Effects of Resilience Training on Resilient Functioning in Chronic Stress Situations among Cadets of the Swiss Armed Forces"

_healthcare, 2023, doi:10.3390/healthcare11091329_

Round 1
Reviewer 1 Report
This manuscript describes a study in which a Resilient Functioning Score was calculated for a group of military cadets and was then correlated with measures of coping strategies, self-efficacy, stress reactivity, and well-being. The authors also used a pre/post design to determine whether or not the Resilient Functioning Score could be improved by a resilience training intervention.
This is, for the most part, a well-written and logically organized manuscript that should be of interest to readers of Healthcare. Once the following changes are made, the paper should be published as soon as conveniently possible.
1. Consider deleting the first paragraph on page 4 since the authors have already established the utility and rationale for the RFS in the previous paragraph.
2. Instead of the paragraph referenced above, the authors should provide some (brief) evidence that infantry training is considered a “chronic stressor.” As the authors know, whether or not something is stressful depends in large part on the individuals’ perception of the event as well as the perception of their preparedness to meet the demands of the event. Some people love challenging training environments while others hate it. What is the general consensus on cadets’ feelings towards infantry training?
3. In the paragraph beginning on line 301, please make it clear that the ANOVA was performed on the RFS’s produced by the PCA.
4. Please delete the fourth and fifth analyses (lines 306-309) because these are largely redundant to the Time X Group ANOVA.
5. In the Conclusions section, please change the first sentence so that it reads: “This study investigated the use of a Resilience Functioning Score, based on the “residuals” method, as a newly direct measurement of resilience.”
6. Please have the paper reviewed and edited by a native speaker of the English language to correct several minor grammatical errors that appear in the text (examples are incorrect punctuation on lines 30 and 47; somewhat awkward wording in the sentence that starts on line 52; usage of the term “its” rather than “his/her” to refer to “subjects” in the sentence starting on line 65; using the term “accurately” rather than “accurate” in the sentence that begins on line 69; etc.).
Once these adjustments have been made, the paper will be suitable for publication. Thank you.
see above
Author Response
We thank Reviewer #1 for her/his helpful comments and suggestions, which helped us to improve the quality of the revision. The point-by-point response is attached as separate file.

Reviewer 2 Report
The manuscript “Effects of resiliency training on resilient functioning in chronic stress situations among cadets of the Swiss Armed Forces” is very well-written, well-structured and informative, presenting an important topic of resiliency functioning and its associations with various psychosocial variables among armed forces.
The findings of the study showed that resiliency functioning was associated with well-being, stress reactivity and higher task-oriented coping. These findings add important information to the literature on resiliency among military- the topic that is essential to be discussed and explored more extensively. The study used various well-validated and known psychosocial instruments which is a big plus. And although the sample size of this study was relatively small, and the results are not generalizable to other populations it is worth publishing, in my opinion.
I would suggest one thing to the authors, however (you could either keep or toss): make the first table (Table 1) a descriptive table where you would stratify by the groups (control vs intervention) and list the means for age and all the other variables of interest for each group – that way readers can see the differences between those two groups prior to the intervention.
Author Response
We thank Reviewer #2 for the care devoted to review our manuscript and for her/his helpful suggestions, which helped us to improve the quality of the revision. The point-by-point response is attached as separate file.
